# Cervical mucus viscoelasticity and sperm velocity are correlated and concentration-dependent *in Vitro*

Matthew R. Markovetz[1]*, Shuhao Wei[2], Chris Celluci[2], Mackenzie Roberts[2], Leo Han[2,3]

**1** Marsico Lung Institute, University of North Carolina at Chapel Hill, Chapel Hill, United States of America, **2** Department of Obstetrics and Gynecology, Oregon Health Sciences University, Portland, United States of America, **3** Division of Reproductive and Developmental Sciences, Oregon National Primate Research Center, Portland, United States of America

* matthew_markovetz@med.unc.edu

## Abstract

### Background

Mucus in the endocervix serves as fertility gatekeeper in the reproductive tract through hormonally regulated changes in biophysical properties. Cervical mucus can thicken to prevent ascension of sperm into the upper reproductive tract or thin to permit fertilization. Current reproductive studies of mucus viscoelastic properties rely on subjective visual appraisal of cervical mucus changes. Our goal was to use particle tracking microrheology (PTMR) to objectively assess cervical mucus visco-elastic properties and associate these measurements with in vitro measures of sperm velocity.

### Methods

Using cervical mucus obtained from rhesus macaques (*Macaca mulatta*) at necropsy, we used to PTMR to measure viscoelasticity ($\eta^*$) under stepwise, serial dilutions. In parallel we measure sperm velocity using custom sperm tracking and analysis workflows.

### Results

We report that both mucus $\eta^*$ and sperm velocity displayed a concentration-dependent behavior, where $\eta^*$ increased as mucus concentration increased, and sperm velocity correspondingly decreased. Viscoelasticity and sperm velocity were strongly negatively correlated ($p < 0.001$).

### Conclusions

PTMR and sperm tracking in mucus provide quantitative measure of viscoelastic mucus changes. PTMR is potentially a method for quantitatively assessing fertility

**Data availability statement:** All relevant data are within the manuscript and its Supporting Information files, as well as a GitHub repository (https://github.com/MMarkovetz/CVM_Paper_Data.git).

**Funding:** This work received funding from the Cystic Fibrosis Foundation (MARKOV22I0, received by MRM), the Bill & Melinda Gates Foundation (INV-024195, received by LH), and the National Institutes of Health (1R01HD115770, received by MRM and LH). The funders had no role in study design, data collection and analysis, decision to publish, or preparation of the manuscript.

**Competing interests:** The authors have declared that no competing interests exist.

potential in the cervix that could be applied to both infertility and contraceptives studies.

## Introduction

Cervical mucus is a critical regulator of female fertility [1] and a natural immune barrier to the upper female genital tract [2]. Mucus biophysical properties fluctuate during the menstrual cycle in response to changes in sex-steroid hormones; mucus can decrease in viscoelasticity during the peri-ovulatory time period to permit sperm ascension into the reproductive tract and increase during other time points to create a thick sperm-impenetrable barrier.[3] In fact, hormonal contraception, in particular progestin-only contraception, relies on this mechanism to maintain high efficacy even when ovulation occurs.

While we understand many of the hormonal regulators and drivers of mucus changes, we have limited tools for measuring observed changes in cervical mucus. The current standard for clinical evaluation of cervical mucus in contraceptive and fertility studies is the cervical mucus score (CMS aka Insler score) [4]. Insler scoring is only semi-quantitative as it is a summation of 5 qualitative assessments of mucus characteristics (ferning, spinnbarkeit (stretchability), viscosity qualitative), cellularity, quantity). While widely used in clinical trials, Insler score is limited in objectivity, reproducibility, and interpretability of the final score as the scoring system has never been directly validated for fertility [4]. Similarly, other measures of mucus that assess the sperm-mucus interaction lack standardized methodologies and are often difficult to interpret.[5] As a result, there remains a need for a robust, quantitative assay of cervical mucus biophysical properties that is related directly to fertility and sperm function.

Prior studies have shown that cervical mucus viscoelasticity and penetrability by sperm are inversely related, with minimum viscosity and maximum penetration occurring during the ovulation period [3,6] Additional studies, primarily in pulmonary mucus, have shown that concentration is the primary determinant of mucus viscoelasticity [7], and several early works on cervical mucus rheology reported that while cervical mucus volume and production is highest during ovulation, its concentration is lowest mid-cycle [6,8] Our hypothesis in this work is that both cervical mucus viscoelasticity and sperm penetrability are mucus concentration-dependent phenomena. Using cervical mucus and sperm obtained from a non-human primate model, we demonstrate that viscoelasticity and sperm motility are inversely related when quantified using particle tracking methodologies.

## Materials and methods

### Non-human primate cervical mucus and sperm collection

Cervical mucus and sperm were obtained from rhesus macaques (*Macaca mulatta*) housed at the Oregon National Primate Research Center (ONPRC) in Portland, Oregon. Mucus was obtained at the time of necropsy from adult female macaques

as previously described [9]. The study was not powered to compare specific cycle phases; therefore, analyses focused on serial dilution within each sample, with each animal serving as its own internal control for concentration-dependent effects. Animals were sacrificed for reasons unrelated to mucus collection via exsanguination after deep anesthesia induced by intravenous pentobarbital (50 mg/kg), consistent with the recommendations of the American Veterinary Medical Association's Panel on Euthanasia. For sperm collection, all males were trained by the ONPRC Behavioral Services Unit for collaborative semen collection by non-sedated electro-ejaculations [10]. Semen samples were collected and allowed to liquefy at 37°C for 30 min before evaluation. The liquid fraction of the sample was washed and resuspended with warm HEPES-buffered Tyrode albumin lactate pyruvate (TALP-Hepes) with bovine serum albumin (BSA) supplemented at 3 mg/ml resuspended in the remaining 1 ml. Ejaculates from at least three individual males were included. Only samples demonstrating ≥50 × 10$^6$ sperm/mL and ≥60% total motility after liquefaction were used. For tracking experiments, sperm were diluted to 1 × 10$^6$ sperm/mL prior to addition to mucus; therefore, baseline semen concentration was not included as a variable in statistical analyses. All work was carried out in strict accordance with the recommendations in the Guide for the Care and Use of Laboratory Animals of the National Institutes of Health. The ONPRC Animal Care and Use Committee reviewed and approved all animal procedures. Fig 1 provides a diagram of general workflow and methodology for the study.

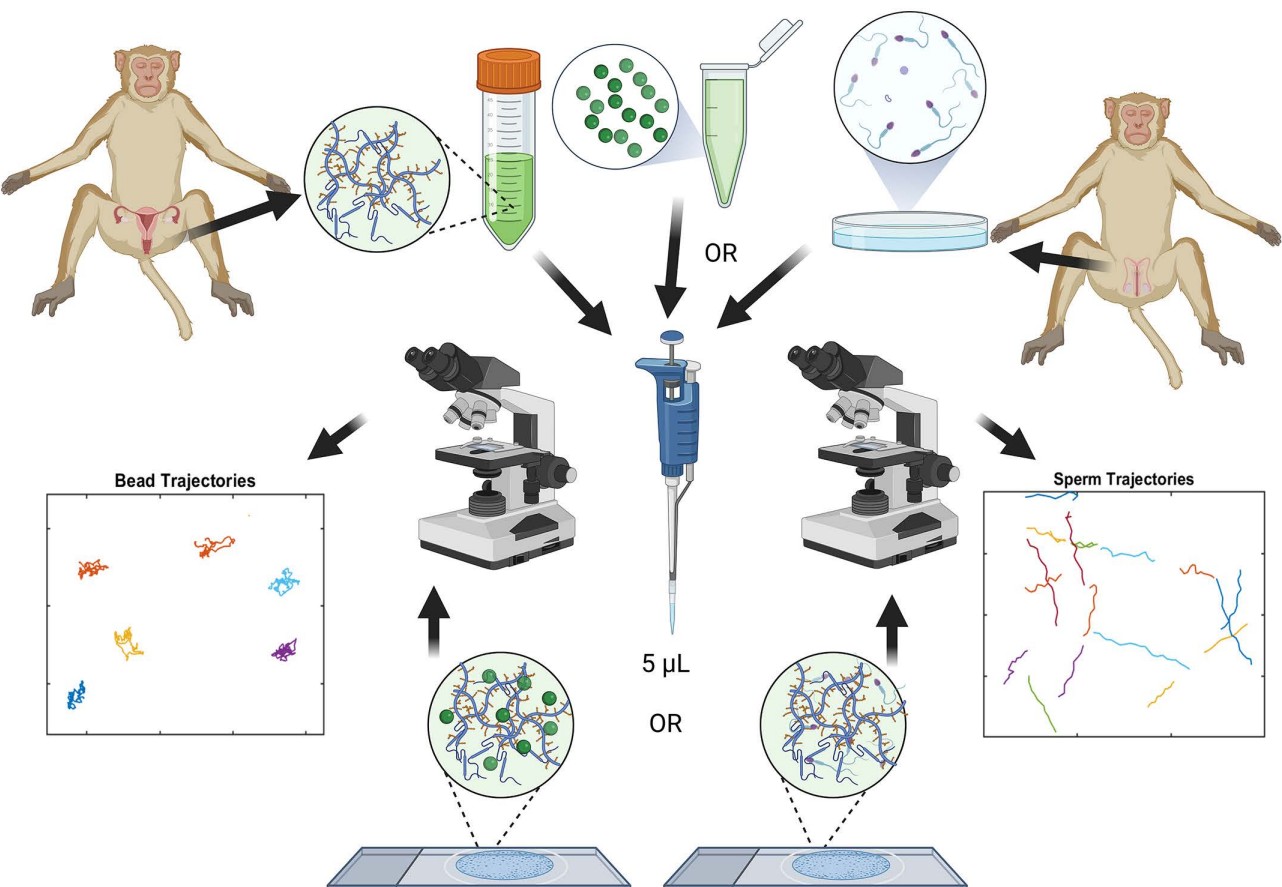

**Fig 1. Schematic of our parallel assessment of sample microrheology and sperm motility in cervical mucus.** Created partially in BioRender. Markovetz, M. (2025) https://BioRender.com/idydjtu.

## Non-human primate handling and humane treatment

Female animals at the ONPRC were group-housed with a vasectomized male for social enrichment. Animals were observed daily for species-specific behaviors, food and water intake, urine and feces production. Abnormalities, if they occur, are reported to the attending veterinarian for evaluation. At the discretion of a veterinarian, animals will receive treatment for treatable conditions, related or unrelated to the experimental protocol. Appropriate treatment may include non-specific oral and/or parenteral fluid therapy, nutritional supplementation consisting of more fruits and special high protein diets, supplemental heat, antipyretics, gastrointestinal protective drugs, specific antimicrobial drug therapy, and the use of analgesics for pain and/or distress. Administration of analgesics concurrent with other treatments will be considered when clinical signs of disease appear as listed above and, if necessary, continued for the duration of the disease state or until an animal is euthanized. Specifically, buprenorphine (0.03 mg/kg IM), or sustained release buprenorphine (0.2 mg/kg SQ) because of its longer duration of action, will be used at frequencies determined appropriate by the veterinarian.

## Particle tracking microrheology and sperm penetration

Rheological measurements of cervical mucus were performed using particle tracking microrheology (PTMR) [11]. Cervical mucus samples were prepared at concentrations ranging from their harvested value and at several diluted values obtained via serial dilution of aliquots into equivalent volumes of Talp-Hepes. Carboxylated, 1 μm, polystyrene beads (Fluospheres, Thermo Fisher, Fremont, CA, USA) were added to mucus samples at a dilution of 1:600. Beaded samples were placed in a sample chamber prepared by cutting out a small square from a strip of Parafilm placed on a glass slide (Fisher). All samples were performed in technical duplicate to account for variability in each aliquot. The chambers were sealed by placing a cover slip over the open portion of the Parafilm. Thermal diffusion of the beads in mucus was recorded for 10 seconds at 30 frames per second using an Insight Gigabit camera (Spot Imaging) on an inverted light microscope (Zeiss AX10). Bead motion was tracked via a custom Python program using the TrackPy package (https://doi.org/10.5281/zenodo.34028). The mean squared displacement of bead diffusion was converted into viscoelastic moduli in accordance with previous methods [12]. Raw data and the Matlab (© 2022 The MathWorks, Natick, MA, USA) files used to process bead and sperm data are provided in a GitHub repository (https://github.com/MMarkovetz/CVM_Paper_Data.git).

Semen was added to parallel aliquots of the same mucus samples prepared and diluted for PTMR at concentration of $10^6$ sperm/ml. Sperm were fluorescently labeled with Hoechst 33342 for 10 minutes at 37 °C in order to improve trackability. Using identical sample chambers as those used in PTMR experiments, videos of sperm motion in mucus were recorded for 5 seconds at 20 frames per second. Sperm motion was tracked using a slightly modified version of the PTMR program to account for the larger size of sperm and their faster, directed motility. The only modifications were to the shape parameters used to identify sperm heads. Code parameter bead_size was increased from 11 to 17, and eccentricity was increased from 0.3 to 0.5, but the underlying algorithm was unaltered. Straight-line velocity was calculated for all sperm trajectories with recorded duration of at least one second. All experiments were performed with technical triplicates.

## Statistics

Linear model assessment was performed using the Matlab (© 2022 The MathWorks, Natick, MA, USA) function *corr*, which automatically computed the correlation coefficient and p-value of the model using a Pearson correlation test. Anderson-Darling tests to determine the appropriate family of distributions for ensemble data in PTMR and sperm were performed using the Matlab function *adtest*. Values reported are ensemble mean values unless otherwise noted. Significance for all tests was determined using an α value of 0.05.

## Results

### Cervical mucus rheology is concentration-dependent

Mucus samples from n = 6 female macaques were obtained at necropsy at unidentified points throughout the menstrual cycle and aliquoted and diluted serially up to dilutions as high as 1:128. The complex viscosity ($\eta^*$) of these samples was measured using PTMR with HEPES buffer ($\eta^* = 10^{-3}$ Pa·s) serving as a reference control. Fig 2A illustrates the general viscoelastic behavior of mucus and sperm motility in mucus as sample dilution increased. Despite some variation due to inherent sample heterogeneity, we identified a trend that decreasing concentration decreased viscosity of the cervical mucus, which is consistent with mucus from other tissues [13]. Due to low sample volumes, absolute concentration in terms of % solids was not determinable. As such, viscosities are reported with respect to relative dilution from the initially obtained sample.

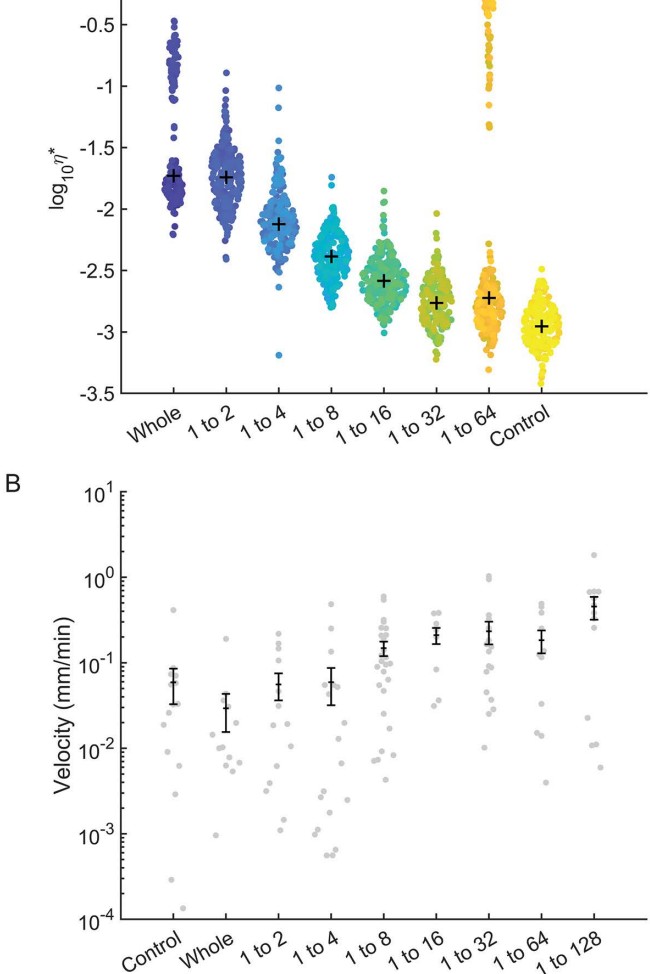

**Fig 2. Representative ensemble behavior observed from (A) PTMR and (B) sperm velocity assessment across a range of serial dilutions in mucus samples obtained from 1 female (out of n = 6 total)** *Macaca mulatta* **and semen from one representative male.** Mean mucus complex viscosity decreased, and sperm velocity increased, as mucus samples became more dilute.

## Sperm velocity within cervical mucus is concentration-dependent

Washed sperm was added to parallel aliquots of the mucus used in PTMR experiments. Sperm tracking in mucus was performed in identical experimental chambers as PTMR as most mucus specimens were too viscous to be loaded into a traditional computer assisted sperm analysis (CASA) device [5]. S1 Video gives an example video of sperm motion in cervical mucus with annotations for successfully tracked spermatozoa. Fig 2B shows the linear velocity of sperm in various concentrations of mucus from a single animal. This sample was representative of overall behavior seen in all animals, where velocity appeared to increase as mucus concentration decreased. Ultimately, decreasing cervical mucus concentration increased sperm velocity scaling with a power law relationship of $\sim c^{0.5}$. Changes in response to dilution were not observed in other CASA parameters such as straightness, linearity, and average lateral head displacement [14].

## Sperm velocity is correlated with mucus viscoelasticity

Because PTMR and sperm motility measurements were performed in parallel, the relationship between mucus rheology and sperm velocity were directly comparable. Fig 3 illustrates that sperm velocity decreased as $\eta^*$ increased in a highly correlated manner ($p < 0.001$). A linear model comparing the median $\log_{10}$ values of velocity and $\eta^*$ was chosen because the log-transformed data most frequently appeared to be normally distributed according to Anderson-Darling tests.

## Discussion

Our studies of particle tracking microrheology and sperm motility in the cervical mucus of rhesus macaques have demonstrated that both mucus viscoelasticity and sperm swimming velocity in that mucus are concentration-dependent. Furthermore, we have shown that viscoelasticity as measured by PTMR is correlated with the velocity of sperm in mucus across the range of concentrations studied. This novel finding suggests that PTMR may be used as a surrogate marker of sperm motility in cervical mucus and therefore may be an indicator of the fertility status of the mucus. Additionally, these studies utilize equipment and techniques that could be easily implemented in a clinical research or office environment and

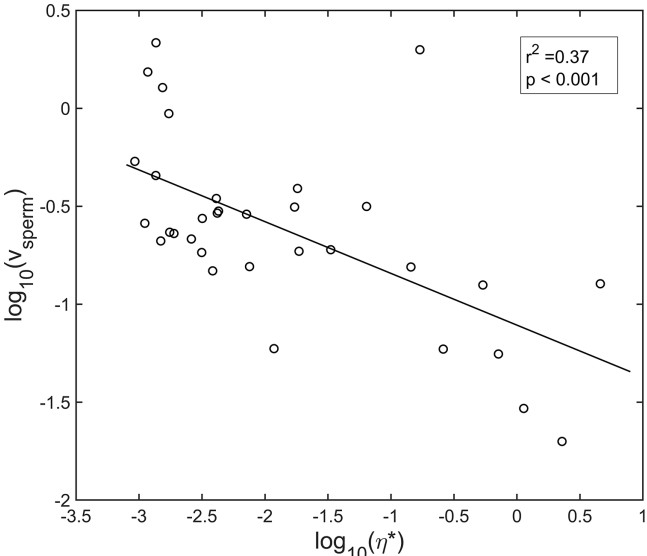

**Fig 3. Mean $\log_{10}$ sperm velocity and $\log_{10}$ complex viscosity (open dots) are inversely correlated using linear regression (Pearson correlation, $p < 0.001$, n = 6 animals).** Dilutions were done in half-fold increments with samples achieving maximal dilutions of at least 1:8 up to 1:128 depending on original sample volume. Sperm velocity and PTMR recordings were done in triplicate for all technical replicates.

better account for the inherent heterogeneity present in both mucus and sperm samples than previously shown. This work represents a step towards more objective and quantitative assessment of fertility attributes of mucus using whole animal samples.

Sperm velocity was highly variable within each sample, illustrating the heterogeneity of both mucus biophysics and sperm motility, but demonstrating that median sperm velocity is reduced by increasing mucus viscosity, our data further supports the existing body of research that designates mucus as a critical regulator of fertility. In the case of existing contraceptives such as progestin-only contraceptives, our findings bolster the notion that a contraceptive effect is likely present through induction of mucus thickening during ovulatory cycles. Our studies may also be useful in understanding fertility changes. Prior to the advent of IUI and IVF, it was thought that up to 15% of women experienced cervical factor infertility, which is infertility related to unfavorable mucus characteristics, even during fertile timepoints. While technologies that physically bypass the cervix have largely rendered this diagnosis moot from a treatment standpoint, there are still clinical scenarios, such as women with cystic fibrosis (CF), where assessment of changes in cervical mucus may help explain the fecundity of an individual [14,16,17].

Our findings, however, may be most useful for studies designed to augment mucus for the purposes of fertility or contraception. Compared to existing clinical appraisal, PTMR could be employed to provide greater rigor and confidence in assessment of biophysical properties that affect sperm function and, ultimately, sperm ascension. Assays like PTMR and sperm tracking serve as probes of these properties in a passive and driven sense, respectively. Measures of sperm penetration represent a logical means to assess fertility-related biophysical properties of cervical mucus. However, existing assays are often limited by poor discrimination power and new technologies such as CASA cannot be used for highly viscous solutions. We have been able to overcome this barrier by employing our modified PTMR technique while reproducing hypothesized and known phenomena: e.g., sperm velocity in the most dilute samples exceeded that of sperm in buffer control, which reflects the known phenomenon that cellular swimmers can be faster in viscoelastic media compared to purely viscous media [15]. Furthermore, our technique may represent a robust alternative to existing CASA methodologies. However, there are limitations to this methodology as well. For example, we must assume that the strong buffer capacity of the mucus itself limits pH changes [7]. Additionally, osmolality affects sperm behavior, but we are unaware of inert reagents that could modify only buffer osmolality without affecting sperm physiology. Meanwhile other rheological evaluations exist. For example, microrheological techniques that evaluate cervical mucus rheology through tracked single particles, sometimes driven by magnets, required much more sophisticated instrumentation [3]. Additionally, single particle techniques are unable to characterize the dramatic heterogeneity of biophysical properties in individual mucus samples. The innovation represented in this work comes from the combination of techniques that assess and relate sperm and bead motion in cervical mucus via techniques that capture the heterogeneity of both sperm motility and mucus rheology using relatively simple equipment (a microscope and camera).

Our study used macaque whole mucus and sperm samples. These studies need to be validated with human cervical mucus and sperm, though the concentration dependence of mucus from other organ systems like the lung and gut has been consistently demonstrated across non-human primate models and in humans, lending confidence to our results in this model [7]. Additionally, our prior studies demonstrate high amounts of proteomic similarity between macaque and human mucus [16]. And while there are other mucosal factors (immunological, pH, microbiological, etc.) that affect fertility, from a purely biophysical perspective, recent studies have shown that mucus concentration is the primary regulator of its viscoelasticity [11,17] Even so, there may be other factors, including changes in the secreted protein milieu that may affect fertility in ways that we do not account for biophysically.

In conclusion, the cervical mucus barrier regulates fertility as its biophysical properties change in response to hormonal changes throughout the cycle. Quantifying the modulation of those changes due to both endogenous and exogenous perturbation is critical to understanding biological mechanisms and finding therapeutics targeting mucus changes. Non-hormonal compounds are of particular interest for both purposes given the off-target effects that are common with

hormone-based drugs. Small molecule drugs that modulate mucus viscosity via ion channels present in both the endo-cervix and other organ systems are one set of likely drug targets [17], [20]. These may represent targets for both contraception and infertility, and their ability to induce biophysical changes in the cervical mucosal barrier can be studied via the techniques developed here.

## Supporting information

**S1 Video. A representative video of sperm motion being tracked.**
(MP4)

## Author contributions

**Conceptualization:** Matthew R. Markovetz, Leo Han.

**Data curation:** Matthew R. Markovetz.

**Formal analysis:** Matthew R. Markovetz, Leo Han.

**Funding acquisition:** Matthew R. Markovetz.

**Investigation:** Matthew R. Markovetz, Shuhao Wei, Chris Celluci, Mackenzie Roberts, Leo Han.

**Methodology:** Matthew R. Markovetz, Leo Han.

**Resources:** Leo Han.

**Software:** Matthew R. Markovetz.

**Supervision:** Leo Han.

**Visualization:** Matthew R. Markovetz.

**Writing – original draft:** Matthew R. Markovetz.

**Writing – review & editing:** Shuhao Wei, Chris Celluci, Mackenzie Roberts, Leo Han.

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
