## [Decision Letter · Decision Letter 0]

29 Jul 2025

Dear Dr. Markovetz,

Thank you for submitting your manuscript to PLOS ONE. After careful consideration, we feel that it has merit but does not fully meet PLOS ONE’s publication criteria as it currently stands. Therefore, we invite you to submit a revised version of the manuscript that addresses the points raised during the review process.

We look forward to receiving your revised manuscript.

Kind regards,

Ayman A Swelum

Academic Editor

PLOS ONE

Journal Requirements:

“MRM, MARKOV22I0, Cystic Fibrosis Foundation, cff.org

LH, INV-024195, Bill & Melinda Gates Foundation, gatesfoundation.org”

4. We note that your Data Availability Statement is currently as follows: All relevant data are within the manuscript and in Supporting Information files.

6. Please ensure that you refer to Figure 1 in your text as, if accepted, production will need this reference to link the reader to the figure.

7. Please include a caption for figure 3.

Additional Editor Comments:

**Please note that one editor commented against publication of your manuscript. Please respond carefully for all reviewers comment.**

Reviewers' comments:

Reviewer's Responses to Questions

**Comments to the Author**

1. Is the manuscript technically sound, and do the data support the conclusions?

Reviewer #1: Yes

Reviewer #2: Partly

Reviewer #3: Yes

2. Has the statistical analysis been performed appropriately and rigorously?

Reviewer #1: Yes

Reviewer #2: I Don't Know

Reviewer #3: N/A

3. Have the authors made all data underlying the findings in their manuscript fully available?

Reviewer #1: Yes

Reviewer #2: Yes

Reviewer #3: Yes

4. Is the manuscript presented in an intelligible fashion and written in standard English?

Reviewer #1: Yes

Reviewer #2: Yes

Reviewer #3: Yes

Reviewer #1: The paper is well written how ever few questions can improve the manuscript more

• What are the possible problems with using cervical mucus from dead macaques, and how might it be different from mucus taken from live animals during their normal cycle?

• Is HEPES buffer a good choice to compare with cervical mucus? Why or why not — does it match the conditions inside the body?

• How could this study help people with fertility problems? What might make it hard to use these results in real life?

• Why did the scientists use log values (log10) instead of the original numbers for speed and thicknes.

• could things like pH or chemicals in the mucus affect sperm speed

Reviewer #2: The manuscript explores the relationship between cervical mucus viscoelasticity and sperm velocity using particle tracking microrheology (PTMR) in rhesus macaques. The current study lacks methodological depth. The data appear sparse. As such, the manuscript falls short of the level of detail and rigor expected for publication. The use of rhesus macaques despite being euthanized for unrelated reasons could have influenced rheological properties is not justified, especially since the conclusions are framed in a human reproductive context. Human cervical mucus and sperm are relatively easy to obtain for in vitro studies across the menstrual cycle.

Reviewer #3: This is an automated report for PONE-D-25-26377. This report was solicited by the PLOS One editorial team and provided by ScreenIT.

ScreenIT is an independent group of scientists developing automated tools that analyze academic papers. A set of automated tools screened your submitted manuscript and provided the report below. Each tool was created by your academic colleagues with the goal of helping authors. The tools look for factors that are important for transparency, rigor and reproducibility, and we hope that the report might help you to improve reporting in your manuscript. Within the report you will find links to more information about the items that the tools check. These links include helpful papers, websites, or videos that explain why the item is important. While our screening tools aim to improve and maintain quality standards they may, on occasion, miss nuances specific to your study type or flag something incorrectly. Each tool has limitations that are described on the ScreenIT website. The tools screen the main file for the paper; they are not able to screen supplements stored in separate files. Please note that the Academic Editor had access to these comments while making a decision on your manuscript. The Academic Editor may ask that issues flagged in this report be addressed. If you would like to provide feedback on the ScreenIT tool, please email the team at ScreenIt@bih-charite.de. If you have questions or concerns about the review process, please contact the PLOS One office at plosone@plos.org.

**Do you want your identity to be public for this peer review?** For information about this choice, including consent withdrawal, please see our For information about this choice, including consent withdrawal, please see our Privacy Policy .

Reviewer #1: No

Reviewer #2: No

Reviewer #3: No

While revising your submission, please upload your figure files to the Preflight Analysis and Conversion Engine (PACE) digital diagnostic tool, https://pacev2.apexcovantage.com/ . PACE helps ensure that figures meet PLOS requirements. To use PACE, you must first register as a user. Registration is free. Then, login and navigate to the UPLOAD tab, where you will find detailed instructions on how to use the tool. If you encounter any issues or have any questions when using PACE, please email PLOS at . PACE helps ensure that figures meet PLOS requirements. To use PACE, you must first register as a user. Registration is free. Then, login and navigate to the UPLOAD tab, where you will find detailed instructions on how to use the tool. If you encounter any issues or have any questions when using PACE, please email PLOS at figures@plos.org . Please note that Supporting Information files do not need this step.. Please note that Supporting Information files do not need this step.

---

## [Author Response · Author response to Decision Letter 1]

12 Sep 2025

Our response to reviewers is attached as a separate document. The unformatted response is here.

Reviewer #1: The paper is well written how ever few questions can improve the manuscript more

• What are the possible problems with using cervical mucus from dead macaques, and how might it be different from mucus taken from live animals during their normal cycle?

The mucus is removed from macaques at the time of necropsy (time of death). In other words, this is more or less a “fresh collection," and we do not expect any differences compared to those taken from an anesthetized live animal.

• Is HEPES buffer a good choice to compare with cervical mucus? Why or why not — does it match the conditions inside the body?

HEPES was chosen because it is a proven (and ideal) buffer for sperm. Our assisted reproductive core (ART core) has optimized sperm buffer based in HEPES for a long time with an extensive publication record and is HEPES is also one of several recommended buffers by andrology guidelines. We added references in our revision. Conceptually, as for a comparator, our focus here is on the viscosity of solution. In this respect, HEPES is like any simple pH-buffering ionic solution in that the viscosity is very low and similar to water (see reference below)>

WHO Laboratory Manual for the Examination and Processing of Human Semen. 6th ed. Geneva: World Health Organization; 2010.

• How could this study help people with fertility problems? What might make it hard to use these results in real life?

Thank you for this question. The thick impenetrable mucus secreted throughout most of the cycle restricts sperm entry into the upper tract for only a few days prior to ovulation. Women who fail to display ovulatory mucus at the time of ovulation carry a diagnosis of ‘cervical factor’ infertility. It is estimated that 5–20% of female infertility results from a ‘cervical factor’. This type of infertility has largely been ignored due to modern infertility treatments like intrauterine insemination allowing clinicians to bypass the cervix with a catheter and the lack of objective tools for measuring mucus changes and correlating them to changes in fertility (ie. effects on sperm). Thus, our study adds a few things to this field: First, it uses an objective measure of viscoelasticity (PTMR), a first in this field as far as we know, to assess cervical mucus samples and this could be used as tool for diagnosis and evaluation of treatment in the future. Second, we directly demonstrate a connection between viscosity of mucus and sperm velocity. While this hypothesized, this direct translational experiment has also not been done and again, further supports the translatability of PTMR measurements. Finally, our use of rhesus also supports the NHP model as platform future studies could be conducted to study interventions (both infertility and contraceptive) targeting the endocervix.

• Why did the scientists use log values (log10) instead of the original numbers for speed and thicknes.

We used a logarithmic representation of the values as it more clearly shows the correlation between speed and viscosity. However, this representation does not alter the underlying values of the data. We suggest that this representation is more easily interpreted than similar presentations using different rheological techniques such as Figure 4 in reference 6 of the manuscript (below).

Wolf, D. P., Blasco, L., Khan, M. A. & Litt, M. Human cervical mucus. IV. Viscoelasticity and sperm penetrability during the ovulatory menstrual cycle. Fertil Steril 30, 163–169 (1978).

• could things like pH or chemicals in the mucus affect sperm speed

Yes. Certainly. For this reason we choose a sperm optimized HEPES based buffer (see prior question)

Reviewer #2: The manuscript explores the relationship between cervical mucus viscoelasticity and sperm velocity using particle tracking microrheology (PTMR) in rhesus macaques. The current study lacks methodological depth. The data appear sparse. As such, the manuscript falls short of the level of detail and rigor expected for publication. The use of rhesus macaques despite being euthanized for unrelated reasons could have influenced rheological properties is not justified, especially since the conclusions are framed in a human reproductive context. Human cervical mucus and sperm are relatively easy to obtain for in vitro studies across the menstrual cycle.

Thank you. As the reviewer notes, human mucus samples can be obtained as we’ve published extensively using clinical trial examining mucus changes. However, this is a translational paper interested in both assessing a novel measure of mucus viscoelastic changes and its correlation with sperm movement. A non human primate model presents a strong and validated experimental platform (see reference below) for these of studies. And while human clinical trials can be a resource for specimen collection, there are also practical considerations that led us to choose NHP mucus, with the most relevant being, the laboratory we conducted these studies takes place within the Oregon National Primate Research Center and we have access to ample fresh NHP specimens and animals. We acknowledge the limitations of translational studies in our limitations section and the usual need for validation studies. We took care not over-extrapolate our findings and keep manuscript length tight to reflect the data presented.

[1]

Han L, Park D, Reddy A, Wilmarth PA, Jensen JT. Comparing endocervical mucus proteome of humans and rhesus macaques. PROTEOMICS – Clinical Applications 2021;15:2100023. https://doi.org/10.1002/prca.202100023

Reviewer #3: This is an automated report for PONE-D-25-26377. This report was solicited by the PLOS One editorial team and provided by ScreenIT.

ScreenIT is an independent group of scientists developing automated tools that analyze academic papers. A set of automated tools screened your submitted manuscript and provided the report below. Each tool was created by your academic colleagues with the goal of helping authors. The tools look for factors that are important for transparency, rigor and reproducibility, and we hope that the report might help you to improve reporting in your manuscript. Within the report you will find links to more information about the items that the tools check. These links include helpful papers, websites, or videos that explain why the item is important. While our screening tools aim to improve and maintain quality standards they may, on occasion, miss nuances specific to your study type or flag something incorrectly. Each tool has limitations that are described on the ScreenIT website. The tools screen the main file for the paper; they are not able to screen supplements stored in separate files. Please note that the Academic Editor had access to these comments while making a decision on your manuscript. The Academic Editor may ask that issues flagged in this report be addressed. If you would like to provide feedback on the ScreenIT tool, please email the team at ScreenIt@bih-charite.de. If you have questions or concerns about the review process, please contact the PLOS One office at plosone@plos.org.

---

## [Decision Letter · Decision Letter 1]

30 Dec 2025

Dear Dr. Markovetz,

Thank you for submitting your manuscript to PLOS ONE. After careful consideration, we feel that it has merit but does not fully meet PLOS ONE’s publication criteria as it currently stands. Therefore, we invite you to submit a revised version of the manuscript that addresses the points raised during the review process.

**ACADEMIC EDITOR: Please respond carefully for the reviewer comments to avoid rejection of your manuscript.**

We look forward to receiving your revised manuscript.

Kind regards,

Ayman A Swelum

Academic Editor

PLOS One

Journal Requirements:

Reviewers' comments:

Reviewer's Responses to Questions

**Comments to the Author**

Reviewer #1: All comments have been addressed

Reviewer #4: (No Response)

2. Is the manuscript technically sound, and do the data support the conclusions?

Reviewer #1: Yes

Reviewer #4: Partly

3. Has the statistical analysis been performed appropriately and rigorously?

Reviewer #1: Yes

Reviewer #4: No

4. Have the authors made all data underlying the findings in their manuscript fully available?

Reviewer #1: Yes

Reviewer #4: No

5. Is the manuscript presented in an intelligible fashion and written in standard English?

Reviewer #1: Yes

Reviewer #4: Yes

Reviewer #1: (No Response)

Reviewer #4: This is in general a well-written manuscript providing new and interesting information. The manuscript sometimes lacks focus and should be improved taking this criticism into account. Some examples are given below.

The authors have also provided satisfying responses to the reviewers’ questions, but they should have added respective changes and explanations to the manuscript. This should be rectified immediately.

In addition, the following concerns must be addressed:

(It is always advisable to provide line numbers for a submitted manuscript to allow for easier review).

In the Material and Methods section, the authors must provide information on the number of semen donors and ejaculates included in the investigation. How was this considered in the statistical analysis. There is also no information if and how semen characteristics were determined. Were there any inclusion/exclusion criteria taken into account?

Cervical mucus was collected at different stages of the reproductive cycle. Please provide information at what stage of the cycle the samples included in the study were collected and if and how this affected viscosity. How were respective differences considered in the study?

“Sperm were labeled with Hoescht…” Please give the respective number and information on the Hoechst staining and correct the name.

“Sperm motion was tracked using a slightly modified version of the PTMR program”. Provide details on the modifications as well as settings used for sperm velocity analysis (e.g. frame settings etc). At what concentration were sperm added to the mucus samples?

What was the pH and osmolality of the (diluted) mucus samples? It is ok that Hepes was used to buffer pH but the pH is not given anywhere. Changes in osmolality will influence sperm velocity.

Information on statistical analysis is not detailed enough.

Some statements in the results section should be moved to the discussion, e.g. “Velocity was highly variable within each sample, illustrating the heterogeneity of both mucus biophysics and sperm motility” and “…, which reflects the known phenomenon that cellular swimmers can be faster in viscoelastic media compared to purely viscous media”. These statements also require additional discussion/interpretation.

The last sentence of the results section (“These results imply that…”) is also an interpretation that is already covered in the discussion. The sentence should be deleted here.

Discussion

I agree with the statement “that PTMR may be used as a surrogate marker of sperm motility… and indicator of the fertility status of the mucus”. This is an interesting and important result. You should consider that upon reversion, PTMR may also be an interesting test for sperm function in men with fertility problems. Please consider this aspect in the discussion.

It is recommended to reduce the paragraph on page 17 (“Our studies may also be useful in understanding fertility changes… cervical mucus hydration as fertility regulator.”) because these aspects were not investigated in this study.

Also, Figure 4 presents an interesting concept, but the contents were not addressed in the present investigation. It should be considered to remove it.

In scientific writing it is improper grammar to begin a sentence with an abbreviation. Please correct this throughout the manuscript.

All figures and tables should “stand alone”, i.e. their legends should provide complete descriptions on what is being presented so that the reader does not have to refer to the manuscript text. Where appropriate, the legend must include information on the species, numbers of individuals or repetitions.

**Do you want your identity to be public for this peer review?** For information about this choice, including consent withdrawal, please see our For information about this choice, including consent withdrawal, please see our Privacy Policy .

Reviewer #1: No

Reviewer #4: No

---

## [Author Response · Author response to Decision Letter 2]

27 Feb 2026

Please see the attached response document for our reviewer response.

---

## [Decision Letter · Decision Letter 2]

23 Mar 2026

Cervical Mucus Viscoelasticity and Sperm Velocity are Correlated and Concentration-dependent In Vitro

PONE-D-25-26377R2

Dear Dr. Markovetz,

We’re pleased to inform you that your manuscript has been judged scientifically suitable for publication and will be formally accepted for publication once it meets all outstanding technical requirements.

Kind regards,

Ayman A Swelum

Academic Editor

PLOS One

Additional Editor Comments (optional):

Reviewers' comments:

Reviewer's Responses to Questions

**Comments to the Author**

Reviewer #4: All comments have been addressed

2. Is the manuscript technically sound, and do the data support the conclusions?

Reviewer #4: (No Response)

3. Has the statistical analysis been performed appropriately and rigorously?

Reviewer #4: (No Response)

4. Have the authors made all data underlying the findings in their manuscript fully available?

Reviewer #4: Yes

5. Is the manuscript presented in an intelligible fashion and written in standard English?

Reviewer #4: Yes

Reviewer #4: The authors provided a very satisfying revision. Both, their responses and the respective changes in the manuscript are much appreciated.

**Do you want your identity to be public for this peer review?** For information about this choice, including consent withdrawal, please see our For information about this choice, including consent withdrawal, please see our Privacy Policy .

Reviewer #4: No

---

## [Editor Report · Acceptance letter]

PONE-D-25-26377R2

PLOS One

Dear Dr. Markovetz,

I'm pleased to inform you that your manuscript has been deemed suitable for publication in PLOS One. Congratulations! Your manuscript is now being handed over to our production team.

Kind regards,

on behalf of

Professor Ayman A Swelum

Academic Editor

PLOS One